# Respiratory Syncytial Virus among People Living with HIV: Is There a Case for Rolling Out Prophylaxis? A Viewpoint Based on a Systematic Review

**DOI:** 10.3390/pathogens13090802

**Published:** 2024-09-16

**Authors:** André Almeida, Raffaele Aliberti, Arianna Aceti, Matteo Boattini

**Affiliations:** 1Department of Internal Medicine 4, Unidade Local de Saúde São José, 1150-199 Lisbon, Portugal; 2NOVA Medical School, 1169-056 Lisbon, Portugal; 3Department of Internal Medicine 2.5, Unidade Local de Saúde São José, 1150-199 Lisbon, Portugal; raffaelejunior.aliberti@gmail.com; 4Department of Medical and Surgical Sciences, University of Bologna, 40126 Bologna, Italy; arianna.aceti2@unibo.it; 5Neonatal Intensive Care Unit, IRCCS AOU Bologna, 40139 Bologna, Italy; 6Department of Public Health and Paediatrics, University of Turin, 10124 Turin, Italy; matteo.boattini@unito.it; 7Lisbon Academic Medical Centre, 1649-028 Lisbon, Portugal

**Keywords:** respiratory syncytial virus, human immunodeficiency virus, vaccination

## Abstract

Respiratory Syncytial Virus (RSV) is responsible for a considerable burden of respiratory disease among children and older adults. Several prophylactic strategies have recently been introduced. We review the available evidence on the interplay between RSV infection and HIV, looking at the specific role of RSV prophylactic strategies in individuals affected by or exposed to HIV. We conducted a systematic review on the association between HIV infection and RSV incidence and severity. We searched in PubMed/MEDLINE for clinical epidemiological studies covering outcomes such as RSV-associated illness, severity, and mortality in individuals affected by or exposed to HIV. A total of 36 studies met the inclusion criteria and were included, the majority conducted in sub-Saharan Africa. There was no compelling evidence suggesting a higher incidence of RSV illness among HIV-infected people. A higher risk of severe disease was consistent among both HIV-positive and HIV-exposed but uninfected (HEU) children. Case fatality rates were also higher for these groups. Evidence on a differing risk among adults was scarce. HIV-positive pregnant women should be given priority for recently approved RSV vaccination, for protection of their newborns. HIV-infected and HEU infants should be considered risk groups for nirsevimab prophylaxis in their first year of life and possibly beyond.

## 1. Introduction

Respiratory Syncytial Virus (RSV) is known to be the main etiological agent of lower respiratory tract infections (LRTIs) in infants and the main cause for hospitalization due to respiratory disease in this age group, mainly for bronchiolitis and pneumonia [1,2]. Beyond its pathogenic implications in pediatric ages, this virus is also increasingly recognized as implicated in acute disease among older adults, presenting a considerable burden of hospital admissions and mortality rates, which are close to those of influenza-driven disease [3,4,5]. Disease severity is aggravated by the presence of respiratory and cardiac comorbidities as well as immunosuppression associated with the bone marrow or a lung transplant [6]. Recently, several prophylactic strategies have been approved and are now being implemented. These include nirsevimab, a monoclonal antibody for universal prophylaxis in infants [7], and two vaccines both approved for use in adults, one of which is also licensed for administration during pregnancy with the aim of infants’ protection [8,9,10]. Concerning vaccines, indications may be extended or strengthened among adults with certain conditions that put them at a higher risk of severe RSV disease, including immune compromise [11,12,13]. These indications are not directly based on evidence acquired during drug trials but rather on the extrapolation of observational data on RSV disease among risk groups.

People living with HIV are known to have a higher risk of unfavorable outcomes for respiratory infections caused by influenza [14,15] and SARS-CoV-2 [16], but data on RSV infections are currently very limited.

In this article, we aim to explore the existing literature on the characteristics of RSV infection among people living with and/or exposed to HIV. Specifically, we sought to determine if they are at higher risk of severe disease and ultimately stronger candidates for vaccination or tailored prophylactic strategies.

## 2. Methods

We carried out a literature search through a systematic review of studies on the RSV infection incidence and severity among HIV patients. A search was conducted in PubMed/MEDLINE combining three different search queries using Boolean operators: “Respiratory Syncytial Virus” OR “RSV” AND “Human Immunodeficiency Virus” OR “HIV”. At the last stage, we searched the reference lists of included studies to look for other potentially relevant studies. Automated duplicate detection was provided by Rayyan software web application [17] and manually confirmed or overruled. After deduplication, title and abstract screening was undertaken against the inclusion/exclusion criteria. The full texts of the selected studies were then reviewed for eligibility.

We included all clinical studies published in peer-reviewed journals that compared the RSV incidence and severity endpoints according to HIV infection and/or exposure. These included ecological, cross-sectional, case–control, and cohort studies, as these are epidemiological frameworks that are equipped to study associations between HIV as an exposure and RSV infection or disease as an outcome. Interventional studies were not expected to be among the retrieved literature as their use would imply a deliberate viral challenge, which carries significant restrictions on ethical grounds. Case reports, case series, and review articles were excluded. Articles based on basic rather than clinical or translational research were excluded. Literature presented in a language other than English was excluded.

Two main reviewers (A. Almeida and R.A.) independently screened all of the literature, selecting, extracting, and excluding reports retrieved during the review process. Conflicts were resolved through collaborative discussion. A data chart was created with a summary of the most relevant aspects present in the included studies. This included the author and year of publication, sampling period, setting, and size, as well as study aims and main findings. Findings of interest were those pertaining to the effect of HIV on RSV outcomes, such as disease incidence, hospitalization, and death. Data charting was carried out primarily by A.A. and checked by all other authors. Data were tabulated separately for hospitalized and community-dwelling patients. Quality appraisal was undertaken using Newcastle–Ottawa Scores. Point estimates of effect measures for studies that used endpoints related to RSV illness severity were graphically represented.

Review items were reported following the Preferred Reporting Items for Systematic Reviews and Meta-Analysis (PRISMA 2020 [18]), whose checklists are presented in Appendix A.

## 3. Results

After an initial identification of 1349 articles meeting the search criteria, 36 were finally included to meet the aim of this review (Figure 1).

Article data including publication and sampling figures, the study type, summary statistics with effect estimates, and quality scoring are presented in Table 1 and Table 2.

Quality assessment through Newcastle–Ottawa Scores is presented in Appendix A. Among the included studies, 27 (75%) were conducted in sub-Saharan Africa (n = 21 in South Africa), 3 in the USA, 2 in Brazil, and 3 in Europe. Moreover, 30 studies included children, 23 of which (64% of the total) were focused exclusively on children. Among these, HIV exposure (to a mother living with HIV) and/or living with HIV were used to differentiate between exposure groups. Nine studies (25%) sampled non-pregnant adults and two pregnant women. The majority of the studies (n = 27; 75%) were conducted in contexts where highly active antiretroviral therapy (HAART) was already made available by local healthcare structures even though this did not translate into a majority effectively taking it [25,30].

There was considerable heterogeneity regarding outcomes, which ranged from the RSV incidence measured by either seroconversion or viral detection [14,21,26,28,29,30,31,34,42,43,45,47,48,50], viral shedding/loads [39,42], death [36,49], hospitalization [3,27,30,35,36,40], or other proxies for disease severity such as the need for oxygen support and length of hospital stay [19,21,22,24,32,33,37,39,44,46].

Looking at the incidence of RSV-related illness, one high-quality study conducted in Kenya found a higher risk among HIV-positive pregnant and postpartum women [48], while a post hoc moderate-quality analysis of influenza vaccine trial data found none [47]. One high-quality study conducted in Brazil where 48% of pregnant women were under HAART did not find a significant difference in RSV seroconversion rates between HIV-exposed and HIV-unexposed infants [43]. HIV-infected children had a higher attack rate of RSV illness in a pre-HAART American cohort during the early 1990s [39]. RSV detection was significantly associated with influenza-like illness in people living with HIV among the 5–24- and 25–44-years age groups in a high-quality study [50]. In a community study looking at the RSV incidence and household transmission, Cohen et al. found no differences in the symptomatic fraction, probability of transmission, or acquisition of infection when comparing HIV-infected with HIV-uninfected [14]. Acute respiratory infection incidence rates were found to be higher among HIV-positive adults in a moderate-quality study in Kenya where no significance testing was performed [46].

One moderate-quality pre-HAART study showed more prolonged RSV viral shedding among HIV-positive children [39], whilst one post-HAART observed no differences among an all-age cohort [14].

Studies analyzing the effect of HIV on RSV illness severity with reported point estimates of association measures are presented in Figure 2. When looking at the severe disease incidence among children, there was high-quality evidence for a higher risk of pneumonia, severe LRTI, and/or hospitalization both among HIV-positive [21,22,25,33,36,51] as well as HIV-exposed but uninfected children (HEU) [22,25,27,30,51]. One study found more frequent bacteremia [22] among those HIV-positive while another found a longer duration of oxygen support among those HIV-exposed or infected [37]. Only one study covering the incidence of severe disease among adults was found (high quality), reporting higher rates among all age groups > 18 years old [23]. One moderate-quality study conducted on an ICU HIV-positive cohort (n = 123) in France did not find differences in RSV detection according to CD4 counts [31].

There was mostly low-quality evidence that among children hospitalized for LRTI, RSV is isolated more frequently from HIV-negative hospitalized children compared to their HIV-positive counterparts [19,21,26,28,29].

Case fatality rates were higher among HIV-exposed children [21,30,36,49]. One countrywide ecological study in South Africa revealed a higher RSV-associated mortality rate for all causes of death among people living with HIV at between 5 and 44 years old [49].

## 4. Discussion

Our review found scarce literature on the interplay between chronic HIV infection and the risk of severe RSV disease among adults, precluding robust conclusions on the use of prophylactic measures in this age group. Nevertheless, as the little evidence available suggests an increased risk, extending the inoculation of approved vaccines may be considered in this population in line with scientific society statements on immunocompromised populations [11,12,13].

As for immunogenicity in people living with HIV, several factors including HIV-related immunological alterations and peculiarities of the target population should be considered. HIV infection is associated with major B-cell immunological defects, including impairments of antibody functions, polyclonal B-cell activation leading to hypergammaglobulinemia, and spontaneous antibody production (Figure 3) [52].

One of the most studied immunological models is that of pregnant women living with HIV and their HEU children, given the attractive opportunity to actively immunize the mother to protect both. A systematic review and meta-analysis, including nine studies from low-and-middle-income countries and three from the USA, showed that HEU children have impaired growth and higher infection-related morbidity/mortality rates, especially in the first six months of life [53]. Women living with HIV have been reported to present with lower immunogenicity for different vaccines, especially in the case of a low CD4+ T-cell count or detectable viral load [54,55]. On the other hand, HIV infection especially in the absence of viral control has been reported to be associated with modifications of the biophysical antibody structure, including glycosylation, also lowering the efficiency of transplacental transfer in pregnant women [56]. This impairment can be further compounded by placental inflammation, induction of maternal hyperglobulinemia through polyclonal B-cell activation, and impairment of the maternal immune response, leading to lower maternal titers of specific antibodies [57]. Both maternal hypergammaglobulinemia and HIV infection have been reported to contribute to low antibody titers at birth and early infections in HEU neonates as shown for tetanus, measles, and Streptococcus pneumoniae [58] or lower specific antibody responses to Haemophilus influenzae type b, Streptococcus pneumoniae, Bordetella pertussis antigens, tetanus toxoid, and hepatitis B surface antigen [59].

Finally, a remarkable reduction in placental transfer of natural RSV-specific antibodies has been reported among pregnant women living with HIV as compared with HIV-uninfected pregnant women [60]. This might explain the considerable evidence elicited in our review that both HIV-infected but also HEU have a higher risk of severe RSV infection, a risk that was consistently observed with several surrogates of severity as endpoints such as hospitalization for severe LRTI [30,40] or death [27].

Therefore, protection of infants following maternal active immunization has been reported to depend on (a) high vaccine-induced antibody titers among pregnant women, (b) efficient transplacental transfer of vaccine-induced antibodies; (c) adequate maternal levels of pathogen-specific IgG when the transfer is most significant—from the last four weeks to full-term pregnancy; (d) a functionally intact placenta; and (e) a term or near-term delivery (Figure 4) [11,61].

Several vaccines administered during pregnancy showed safety and efficacy in the prevention of adverse maternal and neonatal outcomes, although most of the evidence was generated in non-HIV populations [8,54] and limited data on maternal vaccine outcomes are available so far [53].

Regarding RSV in particular, protection against its associated infection is primarily antibody-mediated, with passively acquired neutralizing antibodies that can protect infants in the first months of life [61,62]. Transplacental transfer of RSV antibodies has been defined as highly efficient in mother–infant pairs in rural Nepal, though higher antibody titers showed no protection against earlier or more severe RSV disease in infants [63]. Similar results have been shown in a study on American Indian infants, even if the high rates of RSV severe disease could not be attributed to a failure of protection by the maternal RSV-neutralizing antibodies [64].

Several candidate vaccines have completed phase 3 trials evaluating their safety and efficacy in preventing RSV illness in infants following administration to their pregnant mothers. The recently approved bivalent prefusion RSV vaccine (RSVpreF) demonstrated in its phase 3 trial an efficacy of 81.8% (99.5% CI 40.6 to 96.3) in reducing medically attended severe LRTI and 57.1% (99.5% CI 14.7 to 79.8) in reducing medically attended RSV-associated LRTI within 90 days after birth, though with the latter result not meeting statistical success criteria [8]. The monovalent RSVpreF3-Mat phase 3 trial was stopped early due to safety concerns, as a higher number of preterm births occurred in the intervention arm (relative risk 1.37 95% CI [1.09–1.74]), possibly contributing to a non-significant imbalance of neonatal fatalities (relative risk 2.16; 95% CI, 0.62 to 7.56; *p* = 0.23) [65]. However, accurate measurements of gestational age were lacking for many of the trial participants, possibly contributing to misclassification of age at delivery. Furthermore, no possible mechanisms for an increased risk of preterm delivery were elicited after several post hoc analyses [66]. Efficacy data were nevertheless encouraging, showing a vaccine efficacy of 69.0% (95% credible interval, 33.0 to 87.6) in reducing medically assessed severe RSV-associated lower respiratory tract disease through 6 months of age. For the RSVpreF bivalent vaccine, pregnant persons at an increased risk of preterm delivery were excluded from the phase 2b and phase 3 trials and a small non-significant trend for pre-eclampsia and preterm birth was found. It is worth noting that 72% of preterm births occurred at 36 weeks of gestation. As the benefits seem to substantially outweigh the risks, vaccine recommendations for pregnancy hold, albeit for the period spanning from 32 to 36 weeks of gestational age. The American Center for Disease Control and Prevention reports that post-marketing studies covering the aforementioned adverse events are to be conducted by the manufacturer [67].

Furthermore, a heightened risk of severe disease seemed to persist among HIV-infected children over one year of age in several of the studies covered in this review, rendering the waning effect of maternal immunization less protective over time. Nirsevimab is currently indicated for universal administration in the first year of life at the beginning of the RSV season for children whose mothers were not RSV vaccinated [68,69,70]. This monoclonal antibody, besides being a priority for HIV-exposed infants in their first year of life, could provide an extension of active immunization to HIV-infected children if administered after that period in the same fashion as it is to be considered for other special high-risk populations.

The finding that HIV-negative hospitalized children are more likely to test positive for RSV than their HIV-positive counterparts may seem out of line with the other evidence elicited. The fraction of RSV disease among the overall causes of hospitalization among HIV-positive children is probably lower than that of HIV-negative children, possibly due to the burden of opportunistic disease, which is far less likely among the latter [19,26].

## 5. Conclusions

There is ample evidence, mostly stemming from studies conducted in sub-Saharan Africa, that children who were exposed to HIV-positive mothers carry a significantly higher risk of severe RSV infection. This makes the case for considering prioritizing and extending existing prophylactic strategies to this population. Further data are needed to support a review of the existing recommendations for adult people living with HIV.

## Figures and Tables

**Figure 1 pathogens-13-00802-f001:**
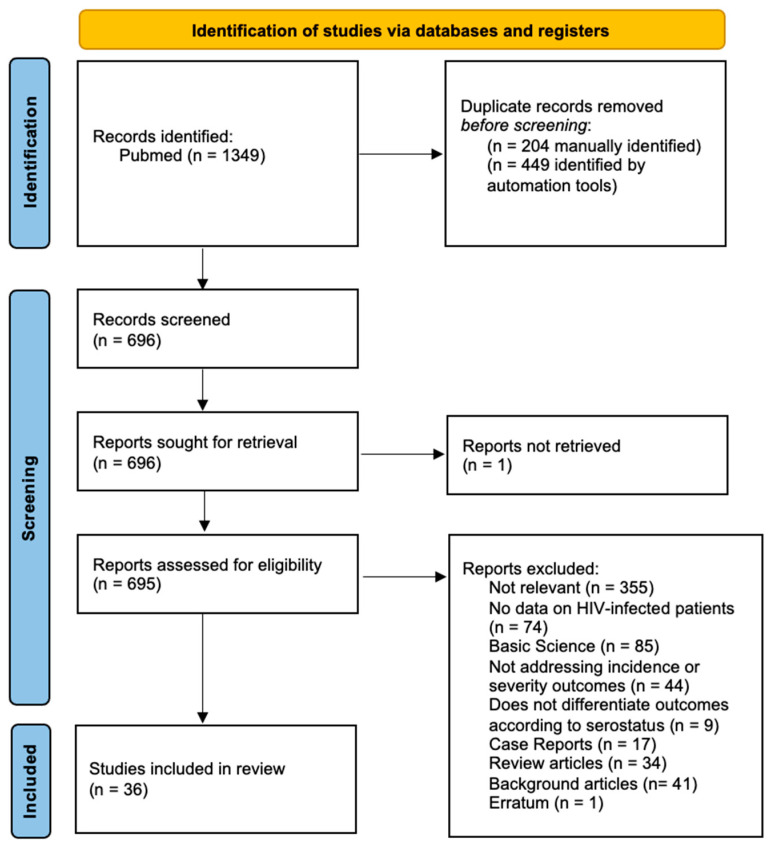
PRISMA (Preferred Reporting Items for Systematic review and Meta-Analysis) Flowchart of included studies.

**Figure 2 pathogens-13-00802-f002:**
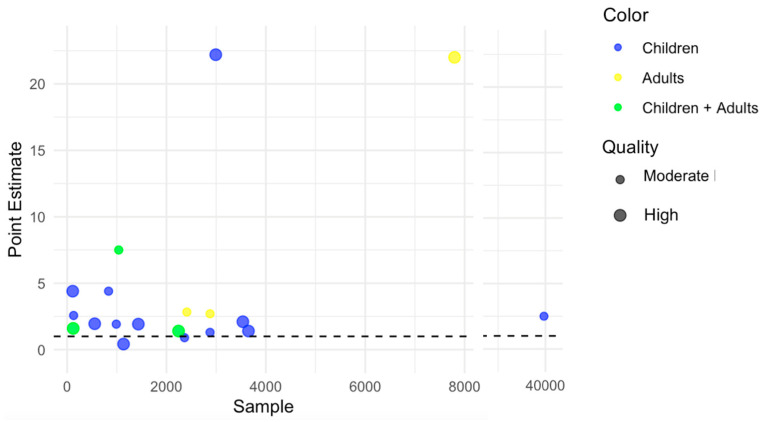
Bubble plot representing point estimates of endpoints used in studies covering RSV severity outcomes.

**Figure 3 pathogens-13-00802-f003:**
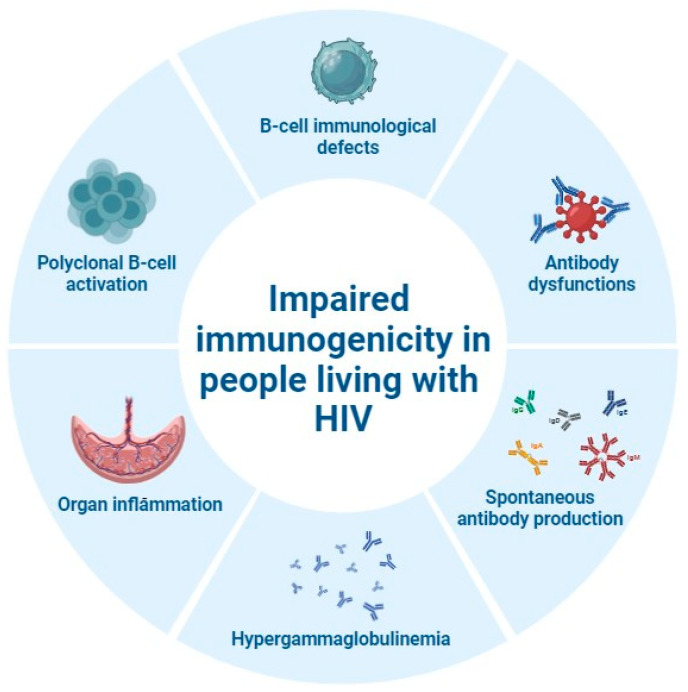
Immunological alterations leading to impaired immunogenicity in people living with HIV (figure created on BioRender.com).

**Figure 4 pathogens-13-00802-f004:**
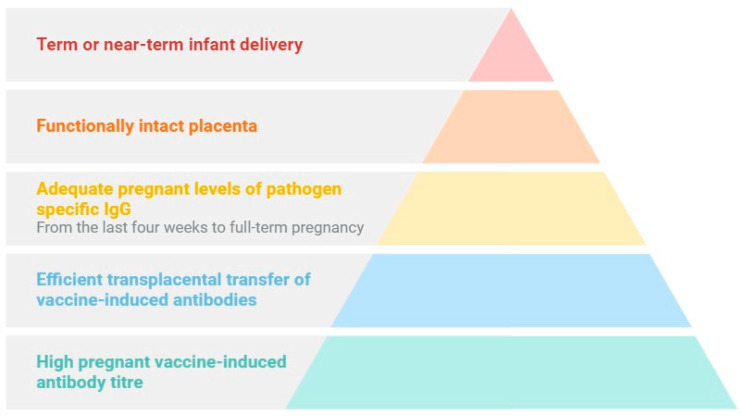
Factors associated with protection of infants following maternal active immunization (figure created on BioRender.com).

**Table 1 pathogens-13-00802-t001:** Main features of studies of hospitalized patients included in the systematic review.

Author and Publication Year	Sampling Period	Country	Population	Study Type	Sample Size (n)	Outcome/Aim	Main Findings	Effect of HIV Exposure	Quality
Owusu 2024 [19]	2017–2020	Ghana	Hospitalized children 3–12 months	Case–control	404	Pneumonia	Non-HIV cases had higher odds of having RSV (7.49, 95% CI: 2.55–21.98)	Decreased risk	high
Liu 2020 [20]	2015–2017	South Africa	Hospitalized children < 3 months with sepsis or LRTI; 3–12 months with LRTI	Cohort	147	Characterize circulating RSV strains	No differences between HIV-exposed and HUU	None	high
Madhi 2000 [21]	1997–1998	South Africa	Hospitalized children 2 months–5 years	Cohort	1434	Viral isolation, disease presentation and LRTI disease burden	Estimated incidence of RSV SLRTI higher among HIV-infected vs. HIV-uninfected (RR 1.92 [1.29–2.83]) RSV isolated less frequently (*p* < 0.001) Higher case fatality rate among all viral LRTIs (*p* < 0.001)	Increased risk	high
Madhi 2001 [22]	1997–1999	South Africa	Hospitalized children < 5 years	Cohort	113	Presentation of RSV-associated LRTI	HIV-infected more likely to present with pneumonia (*p* < 0.002), bacteremia (*p* = 0.003), T > 38 °C, leukocytes > 15,000 (*p* = 0.04), and higher case fatality ratio (RR 4.4) [1.02–18.92]	Increased risk	high
Moyes 2023 [23]	2012–2016	South Africa	Hospitalized (cases) and community-dwelling (controls) children and adults	Case–control	12,048	Non-hospitalized ILI or hospitalized SARI	RSV detection significantly associated with ILI in HIV-positive among 5–24 and 25–44-year age groups (78.1%, 95% CI, 36.5–92.5% and 76.6%, 95% CI 13.6–93.7%, respectively); not significant among SARI cases	Increased risk	high
Madhi 2004 [24]	2001	South Africa	Hospitalized children < 5 years	Cohort	222	Outcomes of nosocomial and community-acquired RSV infection	Risk factors for severe RSV infection more prevalent among HIV-negative (OR 1.92 [95% CI 1.92–3.61]). Non-significant trend for HIV-positive to present with pneumonia; no differences in mortality or length of stay	Increased risk	high
Moyes 2013 [25]	2010–2011	South Africa	Hospitalized children < 5 years	Cohort	2992	ALRTIs requiring hospitalization; hospitalization outcomes	Among all age groups of children, < 5 years higher incidence of RSV-related ALRTIs (RR 3.1–5.6 [2.6–6.4]) HIV-positive more likely to receive oxygen therapy (OR 1.1 [1.0–3.2]), endure a prolonged hospitalization (4.2 [2.0–7.5]), or die (22.2 [4.8–102])	Increased risk	high
Famoroti 2018 [26]	2018	South Africa	Hospitalized children < 5 years	Cross-sectional	2172	Detection of viral respiratory pathogens	Nasopharyngeal swab specimens from HIV-negative patients more frequently associated with RSV detection (*p* = 0.001)	Decreased risk	low
McMorrow 2019 Feb [27]	2011–2016	South Africa	Hospitalized children < 5 years	Cohort	3650	Influenza and RSV-related SARI	HIV exposure associated with increased incidence of RSV-related hospitalization in ages 0–5 months (RR 1.4 [95% CI, 1.3–1.6]); HIV infection associated with RSV-related hospitalization in all age groups	Increased risk	high
Annamalay 2016 Sep [28]	2010–2013	Mozambique	Hospital presenting children < 10 years	Cohort	277	Rhinovirus species among pneumonia inpatients	No significant difference in RSV detection rates between HIV-positive and HIV-negative patients	None	high
Annamalay 2016 Aug [29]	2011–2012	South Africa	Hospitalized children < 2 years	Case–control	208	Describe viral agents among ALRTIs and their interactions with HIV	RSV more frequently isolated from HIV-uninfected patients with ALRTI (*p* = 0.013)	Increased risk	low
Cohen 2016 [30]	2016	South Africa	Hospitalized infants < 6 months	Cohort	3537	Study the epidemiology of LRTI hospitalization in HEU vs. HUU infants	Higher incidence of RSV-related LRTI hospitalization in HEU vs. HUU (IRR 1.4; 95% CI 1.3–1.6); higher case fatality rate of RSV infection (OR 2.1, 95% CI 1.1–3.8)	Increased risk	high
Cohen 2015 [14]	2009–2012	South Africa	Hospitalized children < 5 years	Cohort	8723	Describe LRTI hospitalizations	HIV-infected admitted for ALRTI were less likely to test positive for RSV (*p* < 0.001)	None	moderate
Madhi 2000 [21]	1997–1998	South Africa	Hospitalized children 2 months–5 years	Cohort	990	Estimate the burden of disease and clinical course of viral-associated SLRTI	HIV-infected were less likely to test positive for RSV (*p* < 0.001); estimated incidence higher (RR 1.92 (1.29–2.83)	Increased risk	moderate
Elabbadi 2020 [31]	2011–2017	France	ICU-admitted adults	Cohort	123	Prevalence of respiratory viruses according to the CD4 cell count among patients with ARF	RSV detection not associated with CD4 counts	None	moderate
Chandwani 1990 [32]	1986–1987	USA	Hospital admitted-children (age not specified)	Cohort	28	Describe the clinical course of RSV-related admission	HIV-infected children were more prone to pneumonia and prolonged viral shedding (no statistical testing)	Increased risk	low
O’Callaghan-Gordo 2010 [33]	2006–2007	Mozambique	Hospitalized children < 5 years	Cohort	835	Present surveillance data on the epidemiology of several respiratory viruses associated with clinical pneumonia	Risk of RSV-related pneumonia higher among HIV-infected patients (IRR 2.2–6.5 [no confidence intervals provided])	Increased risk	moderate
Miller 1996 [34]	1994–1995	UK	Adults undergoing bronchoscopy for LRTI work-up	Cohort	44	Viral detection in bronchoalveolar lavage of HIV+-positive patients with respiratory infections	No viruses detected	None	low
Moyes 2017 [24]	2009–2013	South Africa	Hospitalized adults	Cohort	7796	Study the epidemiology of RSV-associated SARI hospitalizations	Among HIV-positive patient age groups, higher risk among 18–64 years (OR 26.3 [6.2–112.1] and 11.4 [2.6–50.0]) compared to older patients and for female sex (OR 2.7 [1.4–5.4]); compared to HIV-uninfected, HIV-infected had longer symptom duration before hospitalization (2 to <5 days aOR 4.4 (95% CI 1.7–11.0) and 5–7 days aOR 4.2 (95% CI 1.6–10.4) compared to <2 days) and higher incidence of RSV-related SARI (RR 14 [11,12,13,14,15,16,17,18,25]	Increased risk	high
Rha 2019 [35]	2009–2014	South Africa	Hospitalized children < 5 years	Cohort	7179	Assess the performance of various case definitions in detecting severe RSV disease	HIV infection reported less commonly in RSV-related SARI cases (aOR 0.4 [0.3–0.5])	None	high
McMorrow 2019 Nov [36]	2011–2016	South Africa	Hospitalized infants < 12 months	Cohort	2243	Magnitude and duration of HIV exposure on RSV hospitalization	HEU had a higher incidence of RSV-related hospitalization compared to HUU until 5 months of age (1.4 [1.4–1.6]); at 6–11 months of age, the difference was not significant	Increased risk	High
Patel 2019 [37]	2012–2016	Botswana	Hospitalized children 1–23 months	Cohort	123	Risk factors for poor outcomes among the subset of children with RSV-ALRI	Longer duration of oxygen support among HE vs. HUU (IRR: 1.63; 95% CI: 1.02–2.61; *p* = 0.04); lack of clinical response did not differ	Increased risk	high

**Table 2 pathogens-13-00802-t002:** Main features of studies of non-hospitalized patients included in the systematic review.

Author and Publication Year	Sampling Period	Country	Population	Study Type	Sample Size (n)	Outcome/Aim	Main Findings	Effect of HIV Exposure	Quality
Smith 2021 [38]	2002–2013	USA	Infants < 56 weeks	Cohort	556	Assess the association between seroconversion and hospitalization	Seroconversion to RSV associated with hospitalization among HEU infants (aRR 1.95 [1.21–3.15]) 4.3% of HEU infants hospitalized due to respiratory infection during first year of life	Increased risk	high
King Jr 1993 [39]	1990–1993	USA	Children < 12 months	Cohort	131	Clinical presentation, viral shedding	More prolonged viral shedding in the HIV-infected vs. the HIV-uninfected (30 vs. 6 days, *p* = 0.038) Attack rate of 49/100 vs. 19/100 child-years and more frequent pneumonia (no statistical testing)	Increased risk	moderate
Madhi 2006 [36]	1998–2004	South Africa	Children > 28 days enrolled in an RCT	Cohort	39,836	Hospitalization and death due to RSV-LRTI	Incidence of hospitalization for RSV-LRTI greater among HIV-positive (OR 2.5 [2.04–3.03]) Case fatality rate also higher (OR 12.7 [3.9–31.4]) No difference in re-admissions	Increased risk	moderate
Wedderburn 2024 [40]	2012–2015	South Africa	Infants < 2 years	Cohort	1136	Hospitalization rates	HEU had higher odds of hospitalization, even though difference not significant for RSV-LRTI	None	high
Cohen 2021 [41]	2016–2018	South Africa	Household members of all ages	Cohort	1116	RSV incidence and household transmission	There were no differences in symptomatic fraction, shedding duration, or probability of transmission or acquisition of infection when comparing HIV-infected with HIV-uninfected	None	high
De Souza Luna 2023 [42]	2005–2013	Brazil	Community-dwelling and hospitalized adults and children	Cohort	1380	Compare RSV detection rates and viral loads	Viral loads and frequency rates not higher among HIV-infected patients, though no statistical tests conducted	None	moderate
Weinberg 2017 [43]	2011–2013	Brazil	Infants < 2 years and their mothers	Cohort	335	Incidence of common childhood respiratory tract infections	No significant difference in RSV seroconversion between HIV-exposed and HUU children	None	high
Peterson 2016 [44]	2011–2014	Malawi	Children 3 months–14 years presenting to the Emergency Department	Cohort	2363	Identify factors associated with clinical severity in LRTIs and coviral clustering	No difference in incidence of RSV-positive SARI between HIV-infected and HIV-uninfected	None	moderate
Mendoza Sanchez 2006 [45]	1989–2003	Spain	Children < 15 years attending an Immunodeficiency Clinic	Cohort	26	Evaluate the occurrence and clinical significance of infections produced by respiratory viruses	RSV was the most commonly isolated virus responsible for LRTIs	None	low
Feikin 2012 [46]	2007–2010	Kenya	Community-dwelling and hospitalized adults and children > 5 years	Case–control	1039	Health-seeking acute respiratory infection (ARI)	Incidence rate of RSV-ARI higher among HIV-positive than HIV-negative adults (0.98 [0.51–1.45] vs. 0.13 [0.59–2.01]), but no significance testing done	Increased risk	moderate
Madhi 2018 [47]	2011–2012	South Africa	Women from midpregnancy until 24 weeks postpartum	Cohort	2410	Incidence of RSV illness	No significant differences in RSV incidence rates among pregnant or post-partum HIV-infected women	None	moderate
Nyawanda 2022 [48]	2015–2019	Kenya	Pregnant and postpartum women and their infants < 24 weeks	Cohort	2877	RSV detection and presentation in mothers and infants	RSV-related ARI incidence was higher among HIV-positive compared to HIV-negative women, both pregnant (*p* = 0.01) and postpartum (*p* = 0.03); no difference in RSV-related signs and symptoms; no difference in incidence among infants up to 12 weeks (HEU vs. HUU)	Increased risk	high
Tempia 2015 [49]	1998–2009	South Africa	Countrywide adults and children > 5 years	Ecological	7536	Estimate deaths attributable to influenza and RSV	RSV-associated mortality rate for all causes of death higher for HIV-positive than HIV-negative persons (aRR 66.1, 95% CI 26.0–167.8)	Increased risk	NA

## Data Availability

No new data were generated.

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
