# Peer review of "Respiratory Syncytial Virus among People Living with HIV: Is There a Case for Rolling Out Prophylaxis? A Viewpoint Based on a Systematic Review"

_pathogens, 2024, doi:10.3390/pathogens13090802_

Round 1

Reviewer 1 Report

Comments and Suggestions for Authors

This is a systematic review of research on the risk of RSV infection in HIV-infected and exposed people, and I think it is well-organized and summarized. In addition, the authors' recommendations for preventing RSV infections are also understandable.

 Minor points:

I think it would be easier to read Table 1 if you rearranged it from vertical to horizontal. Please also adjust the vertical width of supplementary Table 1 to make it easier to read.

 Figure1: PRISMA Flowchart of included studies

âž¡

Figure1: PRISMA(Preferred Reporting Items for Systematic review and Meta-Analysis) Flowchart of included studies

Author Response

Thank your for your comments.

We have amended Table S1 (now S3) as suggested.

We have also clarified the acronym for PRISMA in Figure 1's legend.

Unfortunately, we could not amend Table 1's layout due to formatting requirements.

Reviewer 2 Report

Comments and Suggestions for Authors

The Respiratory Syncytial Virus (RSV) presents a significant public health challenge, particularly for children in South Africa. While new vaccination strategies are being developed, it is essential to evaluate their efficacy in certain complex conditions, such as immunodeficiency. Almeida et al. have investigated the relationship between HIV status and RSV morbidity and severity to assess the potential for administering RSV vaccines to this group. Their systematic review, titled “Respiratory Syncytial Virus among people living with HIV: Is there a case for rolling out prophylaxis? – A viewpoint based on a systematic review,” addresses a timely and important topic. However, the review does have some limitations that should be considered, including the reliance on a single database and the need for clearer methodology and results.

To enhance the manuscript, the methods section could benefit from a more detailed and transparent explanation of the inclusion and exclusion criteria and any limitations (such as restricting the literature to articles in English). While it is mentioned that at least two researchers “screened all the literature,” it is important for high-quality studies that all steps of the investigation, including literature selection, extraction, and exclusion, are conducted independently by at least two researchers and then discussed collaboratively. Furthermore, clarifying the rationale behind the chosen study designs and providing insights into funding sources would strengthen the analysis.

The results from the cited studies exhibit heterogeneity and some contradictions, indicating the need for careful data processing. Although Table 1 is quite informative, its complexity could make interpretation challenging. Organizing the data by population (age, hospitalized versus non-hospitalized, etc.) and study quality, or even dividing it into several tables, could enhance clarity, as the current sorting approach is somewhat unclear. Including a comparative group column would also provide helpful context for terms such as “higher” or “lower.” Additionally, statements not directly pertinent to the primary focus on the increased risk of RSV in HIV-infected or exposed individuals (for example, “RSV-LRTIs were the main cause for hospitalization” for Wedderburn 2024) would be better placed in the main text.

Another important suggestion is the inclusion of graphical representations of the collected data, which would greatly facilitate analysis. For example, pie charts illustrating studies that support or oppose the main thesis across different groups (hospitalized children, non-hospitalized individuals, adults, etc.), with possible inclusions of the coefficients related to sample size and quality scores, could enhance understanding. This could also assist the authors in drawing clearer conclusions.

In the discussion, it would be beneficial to focus more on: 1) the concerns regarding the effectiveness of vaccinating HIV-infected mothers to reduce child morbidity; and 2) the benefit-risk ratio of vaccinating HIV-infected children, even if summarized briefly to address safety considerations.

Lastly, attention to typographical details would enhance the overall quality of the manuscript, including sorting inconsistencies in Table 1 and Table S1, small font size in the author column of Table S1, awkward word hyphenations, and merging terms across different columns in Table 1. Additionally, maintaining consistency in the use of abbreviations (for instance, HEU versus the full term) would improve clarity.

In conclusion, this article offers a valuable analysis of the relationship between HIV status and RSV morbidity in various population groups. With a more detailed methods description, clearer graphical data representation, and a more accurate analysis leading to definite conclusions, it will further contribute to our understanding of this important issue, and it will be worthy of publication.

Author Response

We the authors would like to thank the Reviewer for the article appraisal and for the comments which we found to be very clear and constructive, having provided opportunities to greatly improve our work.

Comment 1: 

To enhance the manuscript, the methods section could benefit from a more detailed and transparent explanation of the inclusion and exclusion criteria and any limitations (such as restricting the literature to articles in English). While it is mentioned that at least two researchers “screened all the literature,” it is important for high-quality studies that all steps of the investigation, including literature selection, extraction, and exclusion, are conducted independently by at least two researchers and then discussed collaboratively. Furthermore, clarifying the rationale behind the chosen study designs and providing insights into funding sources would strengthen the analysis.

Response 1:

Thank you for these comments. We agree with the overall remarks made in this section and we have overhauled it in accordance with the suggestions provided. We also included in the Supplementary Materials two tables depicting compliance with PRISMA 2020 guidelines

"We included all clinical studies published in peer-reviewed journals which compared RSV incidence and severity endpoints according to HIV infection and/or exposure. These included both ecological, cross-sectional, case-control and cohort studies, as these are epidemiological frameworks which are equipped to study associations between HIV as an exposure and RSV infection or disease as an outcome. Interventional studies were not expected to be among the retrieved literature as their use would imply deliberate viral challenge, which carries significant restrictions on ethical grounds. Case reports, case series, and review articles were excluded. Articles based on basic rather than clinical or translational research were excluded. Literature presented in a language other than English was excluded.

(…)

Two main reviewers (AAlmeida and RA) independently screened all of the literature, selecting, extracting and excluding reports retrieved during the review process. Conflicts were resolved through collaborative discussion."

Please see attachment – Table S1 and Table S2

Comment 2:

The results from the cited studies exhibit heterogeneity and some contradictions, indicating the need for careful data processing. Although Table 1 is quite informative, its complexity could make interpretation challenging. Organizing the data by population (age, hospitalized versus non-hospitalized, etc.) and study quality, or even dividing it into several tables, could enhance clarity, as the current sorting approach is somewhat unclear. Including a comparative group column would also provide helpful context for terms such as “higher” or “lower.” Additionally, statements not directly pertinent to the primary focus on the increased risk of RSV in HIV-infected or exposed individuals (for example, “RSV-LRTIs were the main cause for hospitalization” for Wedderburn 2024) would be better placed in the main text.

Response 2:

            Thank you for these comments. We followed your suggestion regarding clarification and subsetting of the information contained in Table 1. We broke it down in two tables, one for hospitalized patients and another for non-hospitalized. We added a column depicting the overall effect of HIV on RSV outcomes. We have removed remarks in the Findings column not directly related to the outcomes of interest.

Please see attachment – Table 1 and Table 2

Comment 3:

Another important suggestion is the inclusion of graphical representations of the collected data, which would greatly facilitate analysis. For example, pie charts illustrating studies that support or oppose the main thesis across different groups (hospitalized children, non-hospitalized individuals, adults, etc.), with possible inclusions of the coefficients related to sample size and quality scores, could enhance understanding. This could also assist the authors in drawing clearer conclusions.

Response 3:          

  Thank you for this suggestion. In line with it, we have created a bubble plot representing studies on severity for which point estimates of effect measures were available.

Please see attachment – Figure 3

Comment 4:

In the discussion, it would be beneficial to focus more on: 1) the concerns regarding the effectiveness of vaccinating HIV-infected mothers to reduce child morbidity; and 2) the benefit-risk ratio of vaccinating HIV-infected children, even if summarized briefly to address safety considerations.

Response 4: 

Thank you for this. We have included a paragraph on safety and effectiveness of RSV vaccines administered to pregnant women. Namely, we decided to include specific concerns regarding safety outcomes which were raised during phase 3 trials.

"The monovalent RSVpreF3-Mat phase 3 trial was stopped early due to safety concerns, as a higher number of preterm births occurred in the intervention arm (relative risk 1.37 95% CI [1.09-1.74]) possibly contributing to a non-significant imbalance of neo-natal fatalities (relative risk 2.16; 95% CI, 0.62 to 7.56; P=0.23) [69]. However, accurate measurements of gestational age were lacking for many of the participants possibly contributing to misclassification of age at delivery. Furthermore, no possible mechanisms for increased risk of preterm delivery were elicited after several post-hoc analyses [70]. Efficacy data were nevertheless encouraging showing vaccine efficacy of 69.0% (95% credible interval, 33.0 to 87.6) in reducing medically assessed severe RSV-associated lower respiratory tract disease through 6 months of age. For the RSVpreF bivalent vaccine, pregnant persons at increased risk for preterm delivery were excluded from the phase 2b and phase 3 trials and a small non-significant trend for pre-eclampsia and preterm birth was found, 72% of which at 36 weeks gestation. As benefits seem to substantially outweigh the risks, the vaccine recommendations hold, albeit for 32 to 36 weeks gestational age. The American Center for Disease Control and Prevention reports that post-marketing studies covering these events are to be conducted by the manufacturer [71]."

Comment 5:

Lastly, attention to typographical details would enhance the overall quality of the manuscript, including sorting inconsistencies in Table 1 and Table S1, small font size in the author column of Table S1, awkward word hyphenations, and merging terms across different columns in Table 1. Additionally, maintaining consistency in the use of abbreviations (for instance, HEU versus the full term) would improve clarity.

Response 5

Thank you very much for pointing these out. We have now corrected them.

 Please see attachment – Table 1, Table 2 and Table S3

Reviewer 3 Report

Comments and Suggestions for Authors

The respiratory syncytial virus (RSV), which belongs to the Pneumovirus genus, is one of the main agents of acute respiratory tract infections that can affect the bronchi and lungs. In most cases, it is responsible for the onset of acute bronchiolitis (inflammation of the bronchioles, increasingly thin branches of the bronchi that penetrate the pulmonary alveoli) and pneumonia, especially in premature babies in the first year of life. Even those who received antibodies from their mothers during pregnancy are vulnerable to infection by the respiratory syncytial virus.

For premature babies, those with congenital heart disease, chronic lung disease and congenital or acquired immunodeficiency are at risk of developing severe forms of the disease. Passive smoking, poorly ventilated and crowded environments and early weaning, as it can affect the strengthening of the child's immune system, are other conditions that favor the manifestation of these infectious conditions.

In this review submitted by André Almeida et al., the authors review the available evidence on the interaction between RSV infection and HIV, looking at the specific role of RSV prophylaxis strategies in HIV-infected or exposed individuals. We conducted a systematic review on the association between HIV infection and RSV incidence and severity. The authors used the PubMed/MEDLINE tool to perform clinical epidemiological study searches covering outcomes such as RSV-associated disease, severity, and mortality in HIV-infected or exposed individuals.

The authors describe that a total of 36 studies met the inclusion criteria and were included, most conducted in sub-Saharan Africa. There was no convincing evidence suggesting a higher incidence of RSV disease among HIV-infected individuals. Higher risk of severe disease was consistent among HIV-positive and HIV-exposed but uninfected (HEU) children. Case fatality rates were also higher for these groups. Evidence on differential risks among adults was scarce. HIV-positive pregnant women should be given priority for the recently approved RSV vaccination to protect their newborns. HIV-infected and HEU-infected infants should be considered at risk for nirsevimab prophylaxis in the first year of life and possibly beyond.

This review is interesting and extremely relevant to the field of study

Author Response

Thank you very much for your comments.

Round 2

Reviewer 2 Report

Comments and Suggestions for Authors

The authors took into consideration of all suggestions.